# Corrosion Behavior of an AISI/SAE Steel Cut by Electropulsing

**DOI:** 10.3390/ma12223782

**Published:** 2019-11-18

**Authors:** Carlos Alberto Montilla, Hernán Alberto González, Valentina Kallewaard, José Luis Tristancho

**Affiliations:** 1Faculty of Technology, Universidad Tecnológica de Pereira, Carrera 27 #10-02 Barrio Álamos, 660005 Pereira, Colombia; 2Department of Mechanical Engineering, Universitat Politècnica de Catalunya, Avda. Víctor balaguer 1, 08800 Barcelona, Spain; hernan.gonzalez@upc.edu; 3Faculty of Mechanical Engineering, Universidad Tecnológica de Pereira, Carrera 27 #10-02 Barrio Álamos, 660005 Pereira, Colombia; valentin@utp.edu.co (V.K.); josetris@utp.edu.co (J.L.T.)

**Keywords:** electropulsing, corrosion rate, turning, microhardness

## Abstract

The effect of electropulsing treatment (EPT) on the surface general corrosion behavior of an AISI/SAE 1045 steel under different machining regimes is studied. In the study, the following variables are alternated: high-speed steel (HSS) vs. hard metal (HM), and with and without the assistance of high-density electropulses. The corrosion rates are determined using comparative studies such as gravimetric analysis, salt spray chamber test, electrochemical polarization curve techniques (PC), and linear polarization resistance (LPR). Differences in surface microhardness were evaluated by applying optical microscopy and planimetric procedures. Specimens subjected to electropulses and turned with HM reported greater reductions of corrosion rates. Changes in corrosion behavior can be explained in terms of grain shape factor *h* variation. The present study demonstrates that electropulsing affects the corrosion behavior of AISI/SAE 1045 steel after the turning process.

## 1. Introduction

In order to minimize negative impacts on the human beings and the environment, two major world-wide tendencies in manufacturing have been gaining prominence: daily processes optimization and the development of modern manufacturing ways to reduce energy consumption, material waste, and costs. Several authors have studied this, for example, with a methodology based on the analytic hierarchy process considering factors like emissions, waste production, hazardous materials [1], or through a method for value stream mapping integrated with life-cycle assessment for sustainable manufacturing [2], or analyzing the role of green and sustainable manufacturing technology [3], or the assessment of environmental impact of all flows consumed during additive manufacturing [4]. For this purpose, a lot of research has focused on different methods to improve both conventional (CP) and non-conventional (NCP) manufacturing processes by using combinations of CP and NCP in what is referred to as hybrid manufacturing processes. Some examples of these methods are, the use of advanced materials in cutting tools and the use of different types of coated cutting tools [5,6], the effects on cutting parameters on tool life, surface roughness and cutting forces with Cubic Boron Nitride (CBN) tools [7], the application of green-lubrication fluids during metal-cutting processes [8,9], novel metal-cutting strategies that minimize the plastic deformation of the workpiece [10,11] as well as the energy consumption (electrically assisted manufacturing processes show lower consumptions [12,13,14]), different comparisons between exergy efficiency definitions for manufacturing processes [15], study of improvements of energy efficiency of manufacturing processes [16], the use of sustainability methods, predictive, and optimization models for sustainable manufacturing [17], examples for energy savings and resources efficiency in practical examples from metal cutting industry [18], and complex cutting tool geometries that improve the interaction between the cutting tool and the workpiece [19,20].

The turning process assisted by high-density electropulsing is a hybrid process that combines conventional turning of metals with a train of electropulses of varying width and frequency [12]. According to results, reported in a previous study on the turning process of AISI/SAE 4140, 1045 and 1020 steels by Sanchez et al. [12], the combined effect of plastic deformation and high-energy electropulsing treatment on the workpiece modifies its surface hardness and decreases both surface roughness and power consumption during machining. Salandro and coworkers [21] denoted that the novel technique of Electrically Assisted Forming processes (EAF) can modify the mechanical properties of the material when an electric field is applied to it during its plastic deformation (tension, compression, or bending tests). It is worth mentioning that material formability is improved due to thermal and non-thermal effects. Currently, EAF processes are still under development, and their industrialization is expected for the near future.

A manufacturing process must guarantee the compliance of the workpiece with several quality criteria. From the mechanical point of view, dimensional requirements, geometrical tolerances, and surface finishes, and from the material engineering perspective, surface stability through time, which can be quantified based on the specific resistance variables (fatigue crack, surface wear, and corrosion). The study of corrosion behavior is really important for the industry, and a large part of any company’s budget, in the industrial sector, is destined for proper maintenance due to corrosion problems, with the aim of preventing and minimizing, as far as possible, the harmful effects of corrosion and extending the useful life of structures and equipment. One way to reduce corrosion is with the use of coatings, a subject in which Fotovvatti et al. [22] have performed a wide review of the different techniques employed. Some authors have made efforts to determine and improve the corrosion behavior of materials after manufacturing or machining. Research on Inconel 718 alloy exposed to corrosion in molten salts [23] found that, only at higher temperatures, the variation of the corrosion environment affects the corrosion rates. Another work on the effects on the surface properties of a finishing process by means of electro-pulsing ultrasonic surface treatment on as-received AISI/SAE 1045 steel, reported that the fabrication of a nanocrystalline surface layer had improved surface roughness and corrosion resistance [24]. Also, a study on the corrosion behavior of titanium alloy strips, after electro-pulsing treatment, determined a correlation between the formation of the surface oxide layer and improvements in the resistance against surface wear and corrosion [25]. Authors such as Zhang et al. [26] have studied the relationship between the surface machining by different operation parameters and the microstructures of the near-surface layers, as well as the corrosion behavior differences, for SA182 Grade 304 Stainless Steel.

To the best knowledge of the authors, there is no similar research on the corrosion behavior of AISI/SAE 1045 steel during the electrically assisted turning process. Its corrosion resistance is known to be poor in comparison to other carbon steels [27], which is the fundamental reason behind the present study. This kind of work could be relevant since this steel has widespread use in the industry due to its good mechanical properties. Accordingly, the present study focuses on the general corrosion behavior of the surface of an AISI/SAE 1045 steel after the electrically assisted turning process. For this purpose, electrochemical and chemical procedures were performed to determine corrosion rates after the aforementioned turning technique: salt spray test (based on gravimetric analysis), polarization curves, and linear polarization resistance (LPR). Finally, differences in corrosion areas and surface microhardness were evaluated by applying optical microscopy and planimetric procedures.

## 2. Materials and Methods

### 2.1. Test Specimens and Tools

The analysis of the effect of electropulsing on the general corrosion behavior resorted to comparative experiments, which included two different cutting tools and variation of both cutting parameters and electrical configuration. The electrically assisted oblique-cutting dry turning process was conducted in a lathe machine model TOZ, ZPS-R5. The specimens and the cutting tool were isolated from the lathe machine by placing them inside polymeric dies. The specimens were mounted between the chuck and the tailstock. A self-made electro-pulse generator supplied positive pulses at its maximum output current intensity of 130 A. Figure 1 shows the electrical system that assisted the lathe machine.

Cold-rolled 1045 carbon steel rods of ½ in of diameter were previously prepared in the lathe machine to match the dimensions of φ12.7 mm × 70 mm of the test specimens. That steel, in a temperate state, has good hardness and toughness, and it is widely used in the industry. The initial material hardness is 108 HRB, and the chemical composition of this steel is listed in Table 1 [28].

Figure 2 shows a simplified schematic of an oblique turning process, the primary shear zone *A_c_* is defined by the OABC plane and the angles and variables involved in the definition of the cutting geometry are principal cutting edge angle *φ*, shear angle *ϕ*, rake angle *γ*, feed *f*, depth of cut *d*, chip thickness *t_c_*, uncut chip thickness *t_o_*, and width of cut *W_c_*.

The classical mechanics of chip removal [29] for the determination of *A_c_*, during the oblique turning process, state that calculations should consider the previously mentioned values for the angles and variables of cutting geometry, determine shear angle *ϕ*, and subsequently *A_c_* can be obtained. From Figure 2, the shear plane area *A_c_* is deduced as the length OA multiplied by the length OC, and considering the geometry of the oblique cut as follows:(1)$$A_{c}=\frac{f\cdot \sin\phi }{\sin\varphi }\cdot W_{c},$$the chip ratio *r_c_* is a characteristic geometric parameter of chip removal processes and for turning is calculated as:(2)$$r_{c}=\frac{f\cdot \sin\phi }{t_{c}},$$from the geometry of the orthogonal cut, the known expression for the shear angle *φ* is deduced:(3)$$\tan\varphi =\frac{\cos\gamma }{r_{c}-\sin\gamma },$$with the known rake angle, principal cutting edge angle (Table 2), and chip dimensions, the shear angle is obtained using Equations (2) and (3). Lastly, the shear plane area *A_c_* is determined using Equation (1).

The high-speed steel (HSS) cutting tool had the following dimensions and geometry: 5/16 in × 5/16m in × 2 ½ in, 8° clearance angle, 14° rake angle, and 1.6 mm nose radius. On the other hand, the hard metal (HM) cutting tool of tungsten, reference SNMG 120404 SH NX2525, had the following geometry: 3° rake angle and the 0.4 mm nose radius. The machining parameters adopted during the metal-cutting tests are shown in Table 2. The spindle speed and feed rate were different for HM and HSS, and they were calculated according to the classical theory of metals; the geometries of both tools were different. In EAF processes, the current pulses circulate through a conductive material submitted to great plastic deformation. In a chip removal process, the zone of greatest plastic deformation is the primary shear area *A_c_*, characterized by a great shear strain rate. Therefore, if an electrical current circulates through *A_c_*, the occurrence of a localized thermal effect happens at the layer being cut. Once the values of *A_c_* and the current provided by the generator are known, it is possible to estimate the current density *J*, which is the current per unit area circulating through the primary shear zone. In this sense, the joule effect can be estimated at the contact area between the cutting tool and the material.

Table 3 reports previous cutting geometry values and the calculations of shear angle *ϕ* and shear plane area *A_c_* under four different test conditions using HSS by electropulsing assistance (HSS-EPT) and without electropulsing assistance (HSS), and HM by electropulsing assistance (HM-EPT) and without electropulsing assistance (HM). Additionally, Table 4 presents the electropulsing parameters (frequency and pulse width), the effective current intensity *I_RMS_*, and the RMS current density *Js.*

### 2.2. Gravimetric Analysis and Salt Spray Test

Since the turned area corresponds to the cylindrical surface of the test specimen, both the gravimetric analysis and salt spray test focused on determining the corrosion rates of these areas, based on the ASTM G1 standard (Standard Practice for Preparing, Cleaning, and Evaluating Corrosion Test Specimens) [30]. Five different test conditions were evaluated: (1) as-received, (2) HSS, (3) HSS-EPT, (4) HM, and (5) HM-EPT. Fourteen specimens were selected for each test condition; the dimensions of the as-received specimens were φ12.7 mm × 120 mm, and the turned specimens had φ10.7 mm × 70 mm. The whole test lasted 168 h, and every 24 h, two specimens from each condition were removed to measure their mass loss.

The confined space of the salt spray chamber was held at temperatures in the range (35 ± 2) °C [31]. The flow of electrolyte (the salt solution was prepared by dissolving 5 ± 1 parts by mass of sodium chloride in 95 parts of water) was guaranteed with compressed air between 12 psi and 25 psi (approximately 0.8 bar to 1.7 bar). After removal from the chamber and before weighting, specimens underwent chemical cleaning (HF 10% alcohol solution) and brass-wire brushing. Then, mass loss was measured with a Vibra balance (220 g range and 0.0001 g resolution). Finally, the corrosion rate *V_c_* was estimated as:(4)$$V_{c}=\frac{K\cdot m}{T\cdot A\cdot \rho },$$where *V_c_* is the corrosion rate expressed in mils per year (mpy), *K* is the constant to obtain mpy (*K* = 3.45 × 10^6^), *m* is the mass loss due to salt chamber exposure and is expressed in grams (g), *T* defines the time in hours (h), *A* is the area exposed to the salt chamber (cm^2^), and *ρ* is the density of specimens (7.85 g/cm^3^).

### 2.3. Polarization Curves (PC) and Linear Polarization Resistance (LPR)

Regarding the electrochemical corrosion behavior of the cross sections of AISI/SAE 1045 specimens, two techniques were applied: polarization curves (PC) and linear polarization resistance (LPR), both in compliance with standard ASTM G 59–97 (Standard Test Method for Conducting Potentiodynamic Polarization Resistance Measurements) [32] and using a potentiostat/galvanostat PG TEKCORR 4.2 USB. In this case, corrosion rates were estimated from:(5)$$V_{c}=0.129\cdot \frac{PE}{\rho }\cdot I_{corr},$$where *V_c_* is the corrosion rate expressed in mils per year (mpy), *PE* is the equivalent weight (27,92), *ρ* is the density of specimens (g/cm^3^), and *I_corr_* is the corrosion current density (μA/cm^2^). *PE* is the inverse of the equivalent number N_EQ_, which is calculated according to Equation (6):(6)$$N_{EQ}=\sum \frac{f_{i}\cdot n_{i}}{a_{i}}$$where
*N_EQ_* is the equivalent number;*f_i_* is the fraction of the alloy element;*n*_i_ is the element’s valence;*a_i_* is the atomic mass.

LPR and PC were applied to the same five test conditions previously described for the salt spray test but focusing on the cross sections of two of the specimens for each test condition. The samples were 10 mm high, and their cross sections (on the order of 1 cm^2^) were polished with sanding paper up to 600 grit and welded to a copper wire to ensure electrical contact. After that, the three-electrode arrangement of these tests included Ag/AgCl as the reference electrode and graphite as the counter electrode. The equipment automatically establishes the open-circuit potential (OCP) as well as the potential range between the cathodic and anodic directions. The scan rate during potentiodynamic polarization was adjusted to 10 mV/s. The specimens were exposed to a corrosive attack in an electrolyte solution of distilled water and NaCl (3.5% in weight). Tafel slopes were obtained at 0 h and 24 h of exposure. The values of corrosion current density (*I_corr_*) and corrosion rates were obtained from the linear polarization resistance (LPR) technique. Lastly, plots of corrosion rates versus time were obtained in Microsoft Excel.

### 2.4. Determination of Corrosion Areas on Cross Sections

In this test, a total of eight specimens were evenly allocated to evaluate the evolution of electrochemical corrosion under four test conditions (HSS, HSS-EPT, HM, HM-EPT). Samples had the same dimensions of those used in Section 2.3 and were previously polished with sanding paper up to 1200 grit and then finished with a Buehler microcloth PSA and 0.3 µm alumina. Subsequently, specimens were exposed to the salt spray chamber under a saline environment of similar characteristics to that used in Section 2.2. Every 15 min, samples were removed from the chamber and cleaned. Areas from their circular crowns were photographed in a Zeiss axio Vert A1 Microscope (Zeiss, Oberkochen, Germany, 50×). The specimens then returned to the chamber until completing 60 min of exposure (four observations). Finally, oxidized areas were measured based on the obtained images by means of software Drafsight 2018 TM [33]. As this test is not based on any existing standard, the authors decided to perform it to have a supplementary assessment.

### 2.5. Micrographic Determination of the Grain Shape Factor and Microhardness

To determine whether electropulsing affects the size of metallographic grains, two methods were applied: planimetric procedures and micro-hardness measurements using a Wilson-Wolpert 600_MRD hardness tester (Orcoyen, Navarra, España). The planimetric procedures of this study were performed in compliance with the ASTM E 112 standard [34] to determine grain shape factor *h* by considering the ratio between the number of vertical and horizontal grain boundary intersections (*Rv* and *Rh*). As a previous step to planimetric procedures, the revealing of the microstructure was made and included polishing of specimens (like Section 2.4), thermal treatment at 200 °C, etching with nital (3%), and microscope observation (Zeiss axio Vert A1). Heat treatment was crucial to increase oxidation in the grain boundary and facilitate the subsequent revealing. Finally, all the *h* shape factors were processed in Minitab 18^®^ software [35] to analyze the differences between group means.

## 3. Results

This section is divided into four subsections: gravimetric analysis and salt spray test, polarization curves (PC) and linear polarization resistance (LPR), corrosion evolution in circular crowns from cross sections (subjected to the salt spray chamber), and microhardness analysis.

### 3.1. Gravimetric Analysis and Salt Spray Test

Firstly, the gravimetric analysis and the salt spray test are analyzed to study the influence of electropulses on corrosion behavior. Accordingly, Figure 3 illustrates the mass losses of the specimens exposed to the salt spray chamber. These data were used in subsequent calculations of corrosion rates.

The results show that the mass losses are greater for the as-received specimens for both cutting tools HM and HSS. Based on corrosion rates from the salt spray chamber, the gravimetric analysis showed EAF-turned specimens have less mass loss, and, therefore, their corrosion resistance is higher in comparison to those conventionally turned. Gravimetric curves showed that after 168 h of exposure to the saline environment, both types of turning processes report similar mass loss. One reasonable explanation could be the probable loss of the outer layer, which was directly affected by electropulsing. Figure 4 presents the results obtained after the salt spray test, where it is shown that among the samples corresponding to the shortest times of exposure in the salt spray chamber, the conventional turning process is responsible for the highest corrosion rates. Throughout the test, the lowest corrosion rates for turned specimens corresponded to those turned with electropulsing assistance regardless of the type of cutting tool. However, this tendency disappeared at the end (after 168 h) when corrosion rates of all turned samples were very similar. This change can be attributed to the probable loss of the outer layer, which not only was directly affected by electropulsing but was also the main difference between specimens at the beginning of the test.

### 3.2. Potentiostatic Polarization Curves (PC) and Linear Polarization Resistance (LPR)

Figure 5 and Figure 6 show the potentiostatic polarization curves obtained from exposing the specimens to corrosion attack at 0 h and after 24 h of attack.

During polarization, anodic and cathodic processes become evident, and the corrosion phenomenon could be governed by a diffusive process since an active-passive evolution was not observed. Figure 5 and Figure 6 show behavior that indicates the presence of general or uniform corrosion on the studied surfaces. The Tafel extrapolation in both the anodic and cathodic zones of the plots allows the calculation of anodic (*βa*) and cathodic (*βc*) slopes and with them, to establish the value of the Tafel constant. These values are listed in Table 5.

Using the LPR technique, the values for corrosion kinematics (corrosion rate), polarization resistance, and corrosion current are obtained. Table 6 reports these values.

In Table 6, looking at the total area and at 0 h of exposure, it was the HSS specimen that presented the highest corrosion rate value (12.58 mpy), while the HM EPT specimen reported the lowest corrosion speed value (1.23 mpy). After 24 h of exposure, it can be seen that the highest corrosion rate value was presented by the as-received specimen (283.84 mpy), and, on the contrary, it was the HM EPT specimen that showed the lowest corrosion speed value (1.86 mpy).

Figure 7 compares the corrosion rates of cross sections subjected to PC and LPR electrochemical techniques. At 0 h, the highest corrosion rates correspond to the conventional turning process, reporting the lowest corrosion resistance for unassisted HSS turning, whereas the highest resistance was reported for HM turning assisted with electropulsing. At 24 h, conventional turning processes are still responsible for the highest corrosion rates. This finding is in strong agreement with the results from the salt spray test. In general, HM had better corrosion resistance than HSS regardless of being assisted with electropulsing or not. PC and LPR electrochemical techniques indicated higher corrosion rates for specimens subjected to unassisted turning.

### 3.3. Corrosion Evolution in Circular Crowns from Cross Sections Subjected to Salt Spray Chamber

The corrosion evolution is evaluated in the cross section of the cut metallic bars that were subjected to the salt spray chamber. As a result, Figure 8 provides details on the evolution of chemical corrosion of samples machined with HM tools. The magnified areas correspond to circular crowns from cross sections of specimens subjected to the salt spray chamber. Samples machined with HSS tools presented similar behaviors.

Table 7 summarizes oxidized areas during this test for each turning process at different exposure times. The most affected areas reported coincide with the highest corrosion rates found in Section 3.1 and 3.2. That is, cases of turning without electropulsing.

### 3.4. Micrographic Determination of the Grain Shape Factor h and Microhardness

Table 8 presents the results for the estimation of the variation in grain shape factor *h* under the previously mentioned test conditions. The estimation was performed by means of a planimetric procedure, according to standard ASTM E 112 [34]. For the test conditions with the HSS tool, an ANOVA analysis with a *p*-value of 0.498 indicates that they belong to the same statistical populations, which means there was not a significant variation of *h*. In the tests of specimens turned with HM, an ANOVA analysis with a *p*-value of 0.007 indicates they belong to different populations, which means that there was significant variation of *h*.

Grain shape factors *h* of the metallographic grains at the edges of the specimens turned with HM-EPT are slightly higher than their counterparts turned without EPT, consequently reducing the number of grain edges or limits, which diminishes the susceptibility to intergranular corrosion. This has been reported for different steels by authors such as [36,37,38].

Microhardness profiles from Figure 9 and Figure 10 compare microhardness between conventional and EPT assisted turning. The lowest microhardness values were found for the nearest points to the surface in both cases of turning with electropulsing assistance. However, this tendency of lower microhardness for EPT assisted cases is more consistent along the data for HM processes. From a depth of 1.5 mm and on, values of microhardness become very similar for all four cases of turning.

Our findings are in strong accordance with two previous studies. Firstly, Groover [29] and Tottem [39] claimed that there is an inversely proportional relationship between metallographic grain size and material hardness. Secondly, Sanchez et al. [12] reported macrohardness reduction in machinery steels turned with electropulsing, probably due to the energy added to the material by the electropulses.

## 4. Conclusions

A study of the effect of turning assisted with high-density electropulsing, on the general corrosion behavior of an AISI/SAE 1045 steel, has been successfully carried out.The study demonstrates that the electropulsing improves the corrosion behavior of AISI/SAE 1045 steel after the turning process, especially with the use of Hard Metal HM cutting tool (1.23 mpy at 0 h and 1.86 mpy at 24 h). Meanwhile, for the HSS tool, similar behaviors were obtained, with values of 9.31 mpy at 0 h and 5.31 mpy at 24 h.The grain shape factor *h* at the edges of specimens turned with HM-EPT were slightly higher than those of their counterparts turned without EPT, consequently reducing the number of grain edges or limits, which diminishes susceptibility to intergranular corrosion.Additionally, the assistance with electropulsing has shown that it reduces surface microhardness of turned workpieces (a reduction of 12.3 HV in the specimens turned with the HM cutting tool, and 6.4 HV in specimens turned with the HSS cutting tool). The lowest microhardness values were found for the nearest point to edges. From a depth of 1.5 mm and on, values of microhardness become very similar to conventional turning with HM tools. This finding sheds light on the effect through the depth of the energy added through electropulsing.Finally, since AISI/SAE 1045 steel has widespread use in the industry, the reported improvements on corrosion resistance points out the convenience of future works on turning processes with HM tools as well as electropulsing assistance.

## Figures and Tables

**Figure 1 materials-12-03782-f001:**
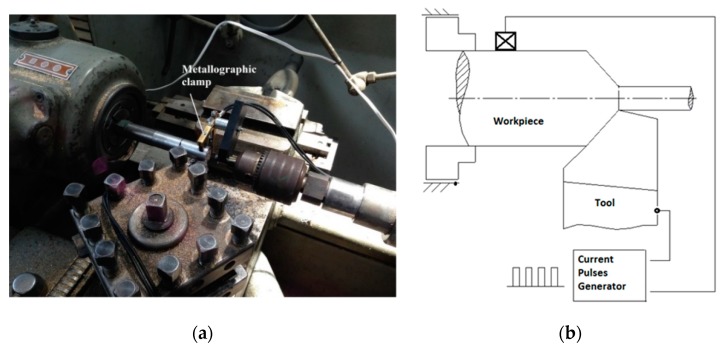
Electrical system and diagram to assist with electropulses. (**a**) Metallographic clamps used; (**b**) Electrical system to assist with electropulses.

**Figure 2 materials-12-03782-f002:**
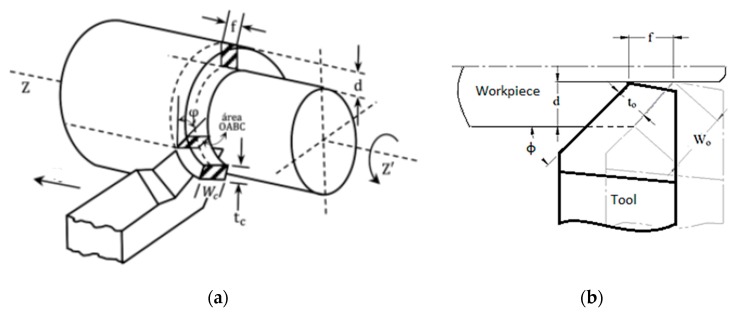
Basic geometry of an oblique turning process. (**a**) OABC plane; (**b**) Some variables in oblique cutting.

**Figure 3 materials-12-03782-f003:**
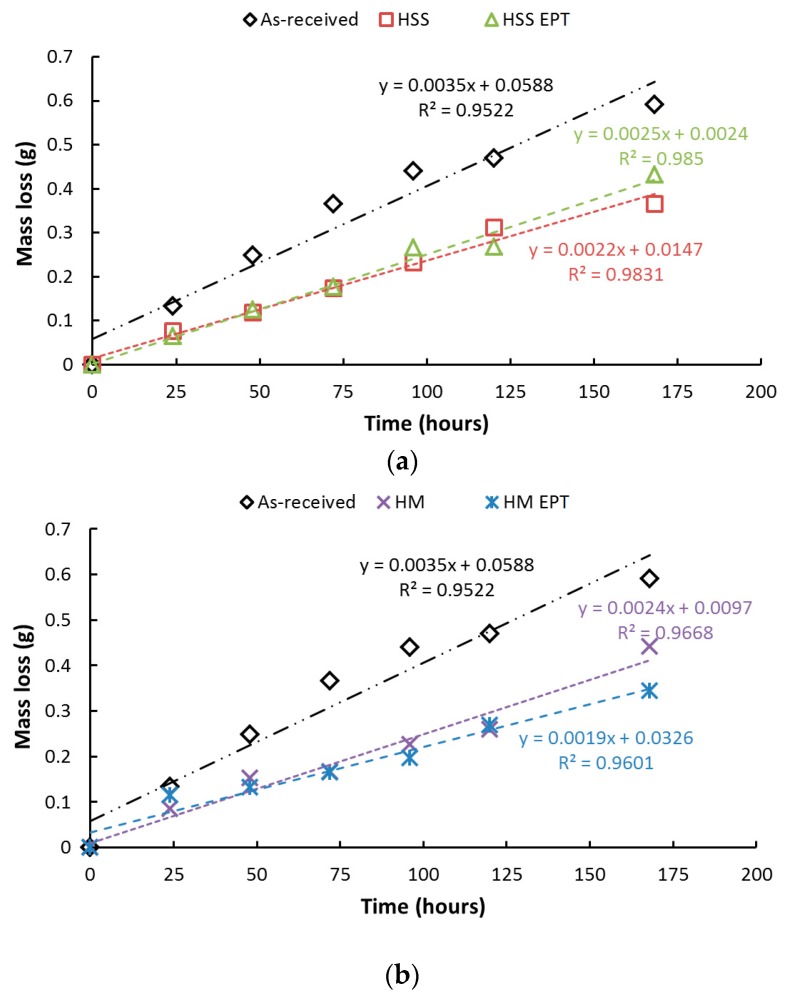
Mass loss during gravimetric analysis: (**a**) high-speed steel (HSS), (**b**) hard metal (HM) with and without electropulsing treatment.

**Figure 4 materials-12-03782-f004:**
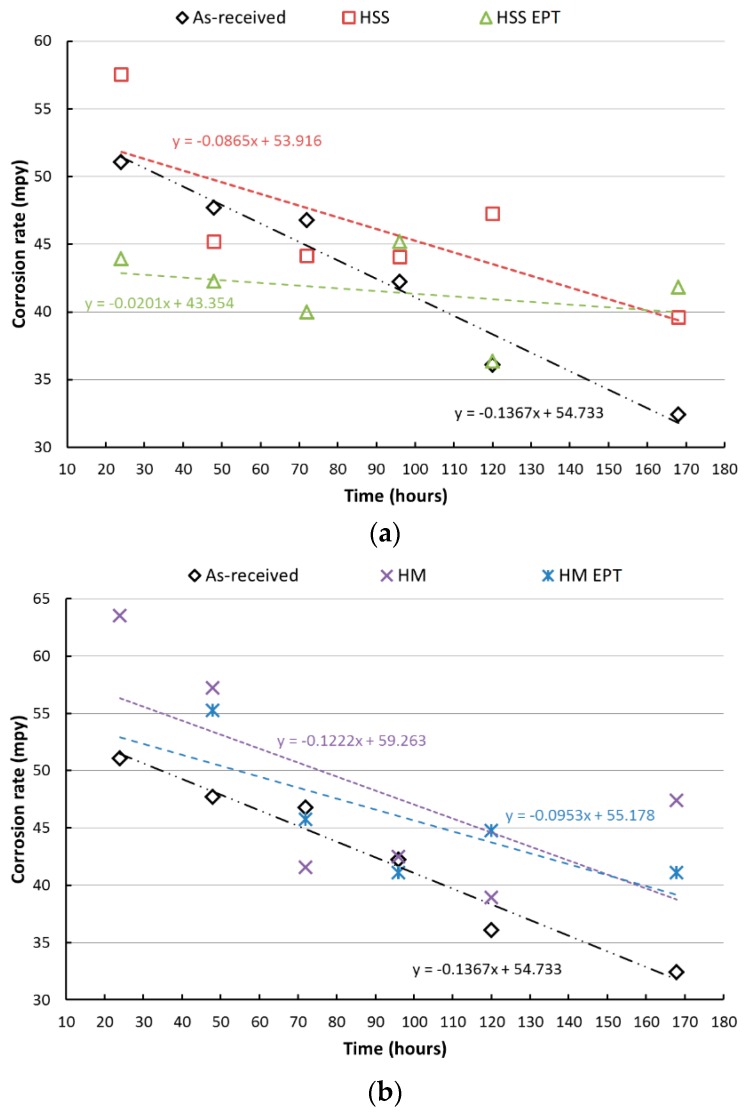
Corrosion rates after the salt spray test. (**a**) HSS tool, (**b**) HM tool with and without electropulsing treatment.

**Figure 5 materials-12-03782-f005:**
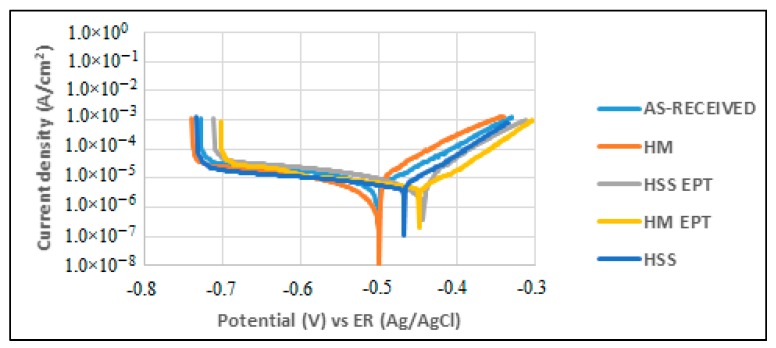
Polarization curves at 0 h.

**Figure 6 materials-12-03782-f006:**
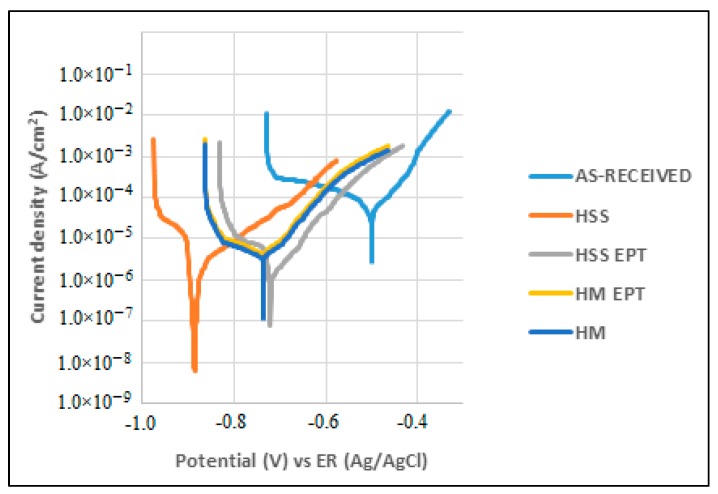
Polarization curves at 24 h.

**Figure 7 materials-12-03782-f007:**
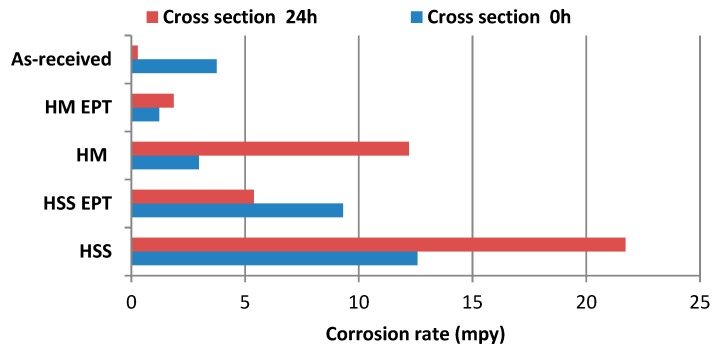
Corrosion rates from cross sections obtained during HSS and HM turning processes.

**Figure 8 materials-12-03782-f008:**
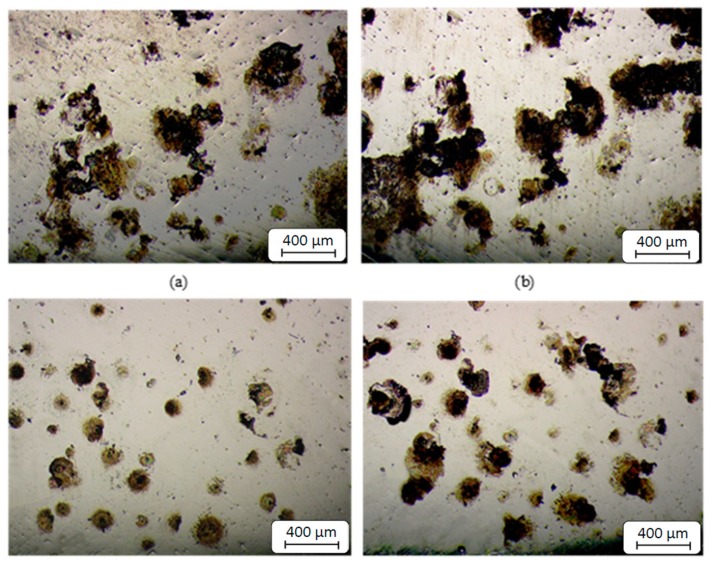
Evolution of corrosion in circular crowns of specimens (50×), (**a**) HM (30 min), (**b**) HM (60 min), (**c**) HM EPT (30 min), (**d**) HM EPT (60 min).

**Figure 9 materials-12-03782-f009:**
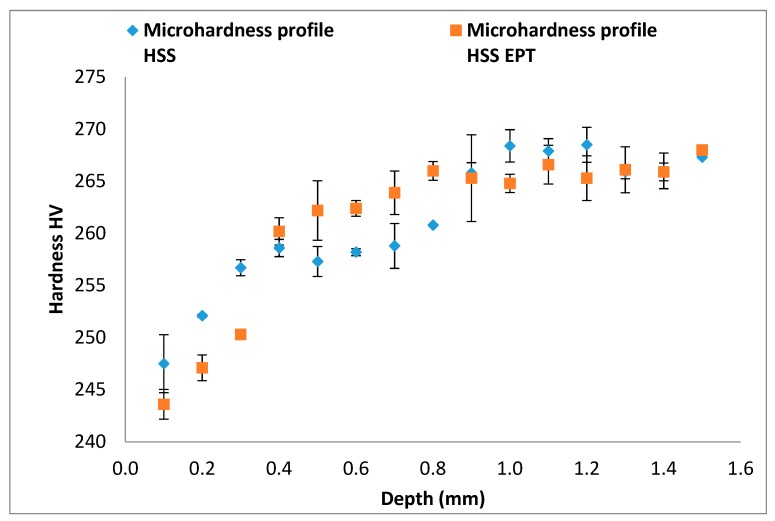
Microhardness profiles of specimens turned with HSS tools.

**Figure 10 materials-12-03782-f010:**
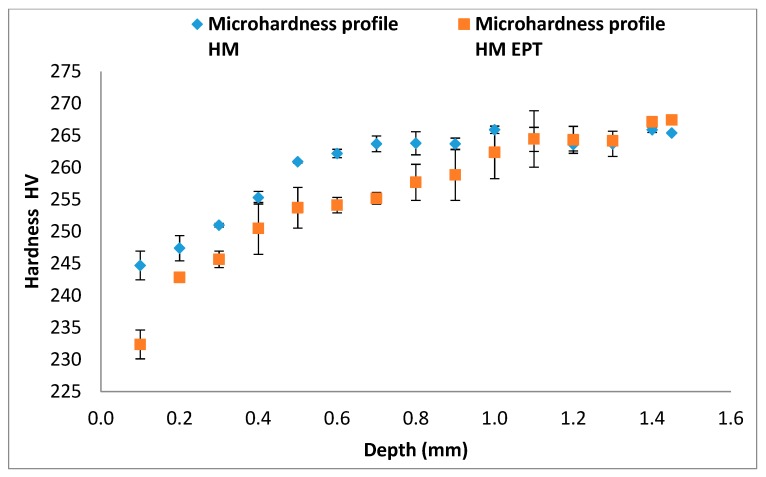
Microhardness profiles of specimens turned with HM tools.

**Table 1 materials-12-03782-t001:** Chemical composition of 1045 carbon steel rods (as-received condition).

Element	C	Mn	Si	S	P	Fe
%	0.45	0.70	0.25	0.007	0.008	Bal.

**Table 2 materials-12-03782-t002:** Metal-cutting parameters.

Tool	Spindle Speed (RPM)	Feed (mm/rev)	Depth of Cut (mm)	Rake Angle *γ* (°)	Principal Cutting Edge Angle (°)
HSS	716	0.138	1	14	40
HM	914	0.174	1	3	30

**Table 3 materials-12-03782-t003:** Cutting geometry values, shear angle ϕ, and shear plane area *A_c_.*

Test Condition	Uncut Chip Thickness to (mm)	Uncut Chip Width *W_o_* (mm)	Chip Thickness *T_c_* (mm)	Chip Width *W_c_* (mm)	Chip Ratio *rc*	Shear Angle *ϕ* (°)	Shear Plane Area *A_c_* (mm^2^)
HSS	0.09	1.56	0.336 ± 0.037	2.116 ± 0.286	0.267	88.5	0.179
HSS-EPT			0.192 ± 0.012	1.924 ± 0.129	0.465	77.1	0.184
HM	0.09	2.00	0.556 ± 0.009	1.563 ± 0.0265	0.156	84.1	0.135
HM-EPT			0.543 ± 0.011	1.521 ± 0.255	0.161	83.8	0.135

**Table 4 materials-12-03782-t004:** Electropulsing parameters during the metal-cutting tests.

Tool	Test Condition	Frequency (Hz)	Pulse Width (µs)	*I_RMS_* (A)	Shear Plane Area *A_c_* (mm^2^)	RMS Current Density *J_s_* (A/mm^2^)
HSS	EPT	300	200	26.93	0.184	146.36
HM	EPT	300	200	26.93	0.135	199.48

**Table 5 materials-12-03782-t005:** Extracted data from the polarization curves on cross sections.

Test Condition	0 h			24 h		
	Cathodic Slope *βc* (V)	Anodic Slopes *βa* (V)	Tafel Constant *β* (V)	Cathodic Slope *βc* (V)	Anodic Slopes *βa* (V)	Tafel Constant *β* (V)
HSS	55.87	252.12	19.86	46.58	121.42	14.62
HM EPT	68.67	57.18	13.55	118.29	65.34	18.28
HSS EPT	144.55	39.38	13.44	58.27	66.51	13.49
HM	245.98	58.5	20.52	361.83	68.75	25.09
As-received	318.84	69.13	24.67	38.73	70.59	10.86

**Table 6 materials-12-03782-t006:** Corrosion current, polarization resistance, and corrosion rate values on cross sections

Test Condition	0 h			24 h		
	Corrosion Current I_corr_ (μA)	Polarization Resistance LPR (Ω)	Corrosion Rate V_corr_ (mpy)	Corrosion Current I_corr_ (μA)	Polarization Resistance LPR (Ω)	Corrosion Rate V_corr_ (mpy)
HSS	97.68	203.3	12.58	168.79	86.61	21.73
HSS EPT	69.64	192.97	9.31	40.35	714.89	5.39
HM	25.59	801.83	2.97	105.3	238.22	12.21
HM EPT	9.04	1500.0	1.23	13.70	1330.0	1.86
As-received	29.15	846.2	3.75	2.2	4930	283.94

**Table 7 materials-12-03782-t007:** Evolution of oxidized areas corresponding to circular crowns of specimens.

Oxidized Areas (µm^2^)				
Time (min)	HSS	HSS-EPT	HM	HM-EPT
15	241,594.8	23,150.5	90,355.2	25,293.3
30	416,886.0	114,623.6	379,033.5	104,153.2
45	674,922.8	426,903.3	575,725.0	242,215.0
60	742,718.2	592,543.5	639,871.2	346,970.7

**Table 8 materials-12-03782-t008:** Variation in grain shape factor *h*, at 200×.

Test Condition	Grain Shape Factor *h*	ANOVA *p*-Value
**HSS**	1.0230 ± 0.1239	0.498
**HSS-EPT**	1.0192 ± 0.1319	
**HM**	1.0980 ± 0.0849	0.007
**HM-EPT**	1.3727 ± 0.1558	

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
