# Peer review of "Corrosion Behavior of an AISI/SAE Steel Cut by Electropulsing"

_materials, 2019, doi:10.3390/ma12223782_

Round 1
Reviewer 1 Report
In my opinion the paper contains essential errors, so I cannot recommend it for publication.
In detail:
Line 103: How was the chemical composition of 1045 steel determined? Moreover in Table 1 is % please specify whether the chemical composition is expressed in wt.% or at.%?
Line 124-125: Equations (with description) used for determining the chip ratio, shear angle and shear plane area should be added.
Line 133(Table 4): On what basis the authors chose pulse width, frequency and current values for EPT?
Line 145: It is an aqueous solution of what?
Line 152: Steel density should be given.
Line 159: Equation (2) is correct only when corrosion current density is in μA/cm2.
Line 160: Equivalent weight of the 1045 steel should be given. In addition, this section should be supplemented with a description of how this value was determined.
Line 171: How were the Tafel slopes determined from Icorr nad Vc? The Authors should explain this statement? Generally, LPR technique, mentioned by the authors, gives a straight line of j=f(E) in narrow range of potentials (usually +-10 mV vs. OCP), thus from this technique, anodic and cathodic Tafel slopes cannot be determined.
Line 172: Graphs shown in the results part of the article are not histograms.
Line 174: Authors interchangeably use words “electrochemical” and “chemical” corrosion, what is incorrect. Mechanism of chemical and electrochemical corrosion is quite different.
Line 188-193: In the article, there is no results from the test declared here.
Line 212: Instead of "corrosion rate" it should be mass loss.
Line 212-213 and 220: What was the thickness of the layer formed using EPT. Moreover, if only a layer is formed by EPT, what is the sense of the electrochemical study of material cross-section (see line 155)?
Line 222 (Fig. 3): There is a lack of point for HM-EPT-24h and R2 for all linear regressions.
Line 228(Fig. 4): How the authors explain vertical lines on the edge of the cathodic branch of Tafel curves? The values on the X and Y axes (order of hundreds of amps and hundreds of thousands of volts) are very unlikely to obtain - the authors should check these values. In addition, instead of ER in X-axis caption, it should be Ag/AgCl.
Line 228 and 230 (Fig. 4 and 5): The Tafel curves shown in Fig. 4 and 5 were registered for differently processed materials. Thus, it is impossible to compare this data and draw appropriate conclusions.
Line 229: Figure caption should be corrected.
Line 233-235: Polarization curves clearly indicate general not local corrosion for the investigated materials. Thus, the statement in lines 233-235 is incorrect.
Line 238(Tab.5): What means “total area ” and “annular area”, how it was estimated, what were their values?
Line 238(Tab.5): How the authors explain the difference in ba, bc and B values obtained for the same material in the same corrosion environment? Unit of ba, bc and B is volt.
Line 240: Why the authors did not present LPR curves. In my opinion, they should be added.
Line 242(Tab.6): How the authors explain that the same material in the same corrosion environment has different corrosion rates.
Line 242(Tab.6): In general there are three basic parameters that describe corrosion behaviour of a material i.e. corrosion current density [A/cm2], corrosion potential [V] and polarization resistance [Ω cm2]. Thus for corrosion current and polarization resistance, shown in Table 6, surface area should be taken into account.
Line 245 and 253-254: The text is inconsistent with Table 6.
Line 295-296: This sentence is unclear and requires more details.
Line 297: Conclusions should be revised taking into account all comments.
Author Response
Line 103 (now 107): How was the chemical composition of 1045 steel determined? Moreover in Table 1 is % please specify whether the chemical composition is expressed in wt.% or at.%?
Table 1 was modified by including the Fe percentage and bibliographic reference [34] where the data sheet from the steel supplier can be found.
Line 124-125 (now line 109 to line 129): Equations (with description) used for determining the chip ratio, shear angle and shear plane area should be added.
The equations, figures and related texts were included (see pages 3 and 4)
Line 133 (Table 4 now line 154): On what basis the authors chose pulse width, frequency and current values for EPT?
According to the work performed by Sánchez and others [12 , 13] the specimens they used in those experiments visually exhibited different corrosion behaviors as the days passed, reason why it was decided to perform the present study, based on the parameters previously determined.
Line 145 (now line 166): It is an aqueous solution of what?
The following description was included: The salt solution was prepared by dissolving (5±1) parts by mass of sodium chlorite in 95 parts of water.
Line 152 (now line 175): Steel density should be given.
The data for density was added inside brackets: 7,85 g/cm³
Line 159 (now line 184): Equation (2) is correct only when corrosion current density is in μA/cm2.
Indeed. This observation was accounted for in line 184: (μA/cm²)
Line 160 (now line 184 to line 191): Equivalent weight of the 1045 steel should be given. In addition, this section should be supplemented with a description of how this value was determined.
The value for PE was added as can be seen in line 183, and the process to obtain it from equation 6 was described between the lines 184 and 191.
Line 171 (now line 199 to line 204): How were the Tafel slopes determined from Icorr nad Vc? The Authors should explain this statement? Generally, LPR technique, mentioned by the authors, gives a straight line of j=f(E) in narrow range of potentials (usually +-10 mV vs. OCP), thus from this technique, anodic and cathodic Tafel slopes cannot be determined.
Because of a transcription error it was written 1 mV/s, but the sampling speed was 10 mV/s. The correction was done in the text (see line 200).
The paragraph between lines 201 and 204 was rewritten, to clarify the process to obtain the TAFEL slopes.
Line 172 (Now line 203): Graphs shown in the results part of the article are not histograms.
The term histogram has been eliminated and was replaced by plots.
Line 174 (Now line 206): Authors interchangeably use words “electrochemical” and “chemical” corrosion, what is incorrect. Mechanism of chemical and electrochemical corrosion is quite different.
We agree with the referee on this subject, however, salt spray chamber tests correspond to chemical corrosion and therefore, in line 206 the term will not be changed, since the concept of electrochemistry wouldn’t apply.
Line 188-193 (Now line 219 to line 222): In the article, there is no results from the test declared here.
The results from the planimetric procedure were included between lines 307 and 317.
Line 212 (Now line 244): Instead of "corrosion rate" it should be mass loss.
Indeed, the term has been changed from corrosion rate to mass loss.
Line 212-213 and 220 (Now lines 244, 245 and 252): What was the thickness of the layer formed using EPT. Moreover, if only a layer is formed by EPT, what is the sense of the electrochemical study of material cross-section (see line 155)?
The depth of the layer formed using EPT wasn’t measured. To infer the depth of said layer, the study in the cylindrical region was compared with a study in the cross section. The paragraph was adjusted accordingly, to clarify the observation performed by the referee.
Line 222 (Fig. 3) (Now line 255, figure 4): There is a lack of point for HM-EPT-24h and R2 for all linear regressions.
Standard ASTM B-117 subsection 10.1, specifies that on Saturdays, Sundays and holydays no observations are performed. The missing point in the regression coincided with a Sunday.
Line 228 (Fig. 4) (Now lines 264 to line 269 – Figure 5): How the authors explain vertical lines on the edge of the cathodic branch of Tafel curves? The values on the X and Y ax(es (order of hundreds of amps and hundreds of thousands of volts) are very unlikely to obtain - the authors should check these values. In addition, instead of ER in X-axis caption, it should be Ag/AgCl.
It has been properly explained between lines 264 and 269.
After a close inspection of the plots presented in the text that was sent, it was observed that they weren’t the final plot results, therefore they have been updated as well as their respective observations.
X-axis caption was corrected according to the observation (Ag/AgCl)
Line 228 and 230 (Fig. 4 and 5) (Now lines 264 to line 269 – Figure 5 and 6): The Tafel curves shown in Fig. 4 and 5 were registered for differently processed materials. Thus, it is impossible to compare this data and draw appropriate conclusions.
We are sorry, but indeed an error occurred when including the figures (they were outdated). The correct figures 5 and 6 were included, which now allow to perform the desired comparisons.
Line 229 (Now line 263): Figure caption should be corrected.
The figure caption has been corrected.
Line 233-235 (Now lines 264 to line 269): Polarization curves clearly indicate general not local corrosion for the investigated materials. Thus, the statement in lines 233-235 is incorrect.
The text was corrected according to what figures 5 and 6 show (lines 264 and 269.)
Line 238 (Tab.5) (Now line 270, Tab. 5): What means “total area ” and “annular area”, how it was estimated, what were their values?
Total area is the full cross section of the specimens. Annular area is a circular crown (see tables 5 and 6) of width R – r = 1.5 mm, assuming it is the region that is most affected by the electropulses.
Tables 5 and 6, and the explanation texts corresponding to annular area or circular crown (lines 253 and 254) were modified, to prevent confusion for the reader.
Line 238 (Tab.5) (Now line 270): How the authors explain the difference in ba, bc and B values obtained for the same material in the same corrosion environment?
Even though it is the same material, it was exposed to different machining processes, with and without the effect of electropulses, therefore the slopes in the anodic and cathodic curves showed different behaviors.
Unit of ba, bc and B is volt.
Table 5 now has the corresponding units.
Line 240: Why the authors did not present LPR curves. In my opinion, they should be added.
Since numeric values were placed in tables 5 and 6, it was considered unnecessary to include the LPR curves.
Line 242 (Tab.6) (Now line 274): How the authors explain that the same material in the same corrosion environment has different corrosion rates.
Although it is the same material, it was exposed to different machining processes, with and without the effect of electropulses, therefore the slopes of the anodic and cathodic curves showed different behaviors.
Line 242 (Tab.6) (Now line 274): In general there are three basic parameters that describe corrosion behaviour of a material i.e. corrosion current density [A/cm2], corrosion potential [V] and polarization resistance [Ω cm2]. Thus for corrosion current and polarization resistance, shown in Table 6, surface area should be taken into account.
Indeed, the exposed Surface area was considered when operating the potentiostat-galvanostat equipment, but was considered unnecessary to include said data in table 6.
Line 245 (now 277) and 253-254: The text is inconsistent with Table 6.
Indeed line 277 had a typing error. The term EPT was suppressed in line 277.
Tables 5 and 6, and the explanation texts corresponding to annular area or circular crown (lines 253 and 254) were modified, to prevent confusion for the reader.
Line 295-296 (Now lines 332 to 334): This sentence is unclear and requires more details.
The affirmation stated in lines 332 and 334 is consistent with what was reported by Sánchez and what is shown in figures 9 and 10, in which the reduction of microhardness at the edges can be seen on the specimens turned with the assistance of electropulses.
Line 297: Conclusions should be revised taking into account all comments.
Conclusions were revised accordingly.
Reviewer 2 Report
Thank you for the great effort you put on this manuscript. I would like to have some comments which I believe strengthen your work:
the introduction is not up-to-date. the introduction of a paper includes the most notable publications and the most recent ones in order to justify that your work is not a repetition of the previous works. so, it would be great if you could update the introduction section with some more recent papers. you should mention why you did not use coatings for reducing the corrosion resistance of the samples since coatings are the most common ways of reducing corrosion rate. the following papers include useful information on coating processes for corrosion resistance enhancement, the effect of machining processes on corrosion rate, etc. it is strongly recommended you include these more recent publications in the introduction section and add a paragraph on the mentioned discussions. of course, you can find more publications and add them in addition to the recommended papers:
https://www.mdpi.com/2504-4494/3/1/28
https://www.sciencedirect.com/science/article/pii/S1359645418309224
https://link.springer.com/article/10.1007/s12666-018-1467-9
figure 1.b is of low quality. please reproduce the figure or replace it with a higher quality image. the table 1 is incomplete. please specify the other elements in the materials you used in your research. for example, if you used steel, you should add Fe element as "balance" if there are no other elements available in your composition analysis. in section 3.2. please indicate what type of the TAFEL curves. Also, please add more discussion on figures 4 and 5. the corrosion curves are complicated and need a deeper discussion. there are not enough information on these two figures. it is strongly recommended you use the following paper and use their discussion on corrosion curves and add more discussion in section 3.2.
https://link.springer.com/article/10.1007/s10853-019-03375-1
please add more discussion on figure 6. why they are different? please make the conclusion in bulletpoints to make it easier to understand for readers.
Author Response
The introduction is not up-to-date. the introduction of a paper includes the most notable publications and the most recent ones in order to justify that your work is not a repetition of the previous works. so, it would be great if you could update the introduction section with some more recent papers. you should mention why you did not use coatings for reducing the corrosion resistance of the samples since coatings are the most common ways of reducing corrosion rate. the following papers include useful information on coating processes for corrosion resistance enhancement, the effect of machining processes on corrosion rate, etc. it is strongly recommended you include these more recent publications in the introduction section and add a paragraph on the mentioned discussions. of course, you can find more publications and add them in addition to the recommended papers:
https://www.mdpi.com/2504-4494/3/1/28
https://www.sciencedirect.com/science/article/pii/S1359645418309224
https://link.springer.com/article/10.1007/s12666-018-1467-9
A revision to the introduction was performed. The recommendations were considered, and 2 more recent references were included.
Figure 1.b is of low quality. please reproduce the figure or replace it with a higher quality image
Figure 1.b was replaced with a better quality one.
The table 1 is incomplete. please specify the other elements in the materials you used in your research. for example, if you used steel, you should add Fe element as "balance" if there are no other elements available in your composition analysis.
The recommendation was followed, and Fe was included
The recommendation was considered, and the balance was included with its respective percentage.
in section 3.2. please indicate what type of the TAFEL curves.
They are of the potentiostatic type.
Also, please add more discussion on figures 4 and 5. the corrosion curves are complicated and need a deeper discussion. there are not enough information on these two figures. it is Strongly recommended you use the following paper and use their discussion on corrosion curves and add more discussion in section 3.2.
https://link.springer.com/article/10.1007/s10853-019-03375-1
We have analyzed the recommended paper. It is about a biomedical application with a Nickel-Titanium (NiTi) alloy, obtained from a process of additive manufacturing and with a micro-arc oxidation coating (MAO). This is very dissimilar to the material and processed we worked with. Sincerely we didn’t find a way to correlate them.
Please add more discussion on figure 6. why they are different?
Based on the applied methods and the results obtained, we honestly don’t have additional arguments to add to what is already described from line 283 to line 290.
Please make the conclusion in bulletpoints to make it easier to understand for readers.
This was performed and the number of conclusions in the text was increased
Reviewer 3 Report
The effect of electropulsing treatment on the corrosion behaviour of high-speed steel and hard metal was presented in this paper. I am confused if this is an experimental or review paper because the paper was categorised as review paper. However, authors performed hardness and corrosion testing on AISI/SAE steel by electropulsing. I recommend the paper to be rejected because of the low significance of the content and serious flaws. My comments are listed below.
The experimental section consists of four pages long which is too lengthy. The experimental section should be concise, and it is suggested to move some parts to supporting information. In section 3.1, “Gravimetric curves showed that after 168 h of exposure to saline environment both types of turning processes report similar mass loss.” Do you have any evidence of probable loss of outer layer that caused by electropulsing? Please provide reference on this explanation. Please make sure all figures to be presented consistently. The font size and font types are different for all figures throughout the paper. Similarly, the polarisation data for different samples should be consistent in Table 5 and Table 6. Table 5 starts with HSS and Table 6 starts with as-received sample. It makes the readers difficult to follow. What is the meaning of “total area” in the heading? What is the reason for the corrosion rate of as-received sample is lower than the corrosion rate of HM EPT and HSS EPT? Please provide more discussion on the results obtained for corrosion testing. Please explain about ANOVA analysis because it is not described in experimental section. The headings on the top row in Table 8 are not written in English. Is the shape factor h for HM-EPT 1.3727 or 1,3727? Please provide references to support your explanation on “Shape factors h of the metallographic grains at the edges of the specimens turned with HM-EPT are slightly higher than their counterparts turned without EPT, consequently reducing the number or grain edges or limits, which diminishes the susceptibility to intergranular corrosion.” In the abstract, authors claimed that “changes in corrosion behaviour can be explained in terms of metallographic micro hardness variation”. However, I do not find any relevant information explaining this mechanism in the discussion.
Author Response
The effect of electropulsing treatment on the corrosion behaviour of high-speed steel and hard metal was presented in this paper. I am confused if this is an experimental or review paper because the paper was categorised as review paper. However, authors performed hardness and corrosion testing on AISI/SAE steel by electropulsing.
Indeed, it is an experimental paper (type: article).
The experimental section consists of four pages long which is too lengthy. The experimental section should be concise, and it is suggested to move some parts to supporting information.
We agree with you, the experimental section is lengthy, but it is so because in the first round of reviews some of the reviewers required to detail more some aspects (those concerning about machining), and it caused the experimental section to increase.
In section 3.1, “Gravimetric curves showed that after 168 h of exposure to saline environment both types of turning processes report similar mass loss.” Do you have any evidence of probable loss of outer layer that caused by electropulsing? Please provide reference on this explanation.
We have no evidence about it. In that sense, the investigation will continue, in order to determine the presence of the outer layer and its probable loss caused by electropulsing.
Please make sure all figures to be presented consistently. The font size and font types are different for all figures throughout the paper.
The figures have been revised and the same font type was used. The figures were revised following the journal's guideline for figures and tables:
All Figures, Schemes and Tables should be inserted into the main text close to their first citation and must be numbered following their number of appearance (Figure 1, Scheme I, Figure 2, Scheme II, Table 1, ). All Figures, Schemes and Tables should have a short explanatory title and caption. All table columns should have an explanatory heading. To facilitate the copy-editing of larger tables, smaller fonts may be used, but no less than 8 pt. in size. Authors should use the Table option of Microsoft Word to create tables.
Similarly, the polarisation data for different samples should be consistent in Table 5 and Table 6. Table 5 starts with HSS and Table 6 starts with as-received sample. It makes the readers difficult to follow.
You are right. The tables have been modified.
What is the meaning of “total area” in the heading? What is the reason for the corrosion rate of as-received sample is lower than the corrosion rate of HM EPT and HSS EPT?
The meaning of “total area” is the cross section of specimens. In order to avoid any confusion, the term “total area” was removed from the paper.
Please provide more discussion on the results obtained for corrosion testing.
In order to provide more discussion, we consider necessary to develop the next phase of the investigation.
Please explain about ANOVA analysis because it is not described in experimental section.
An Analysis of Variance or ANOVA test is a way to find out if survey or experiment results are significant. It is a statistical tool for general use, therefore we don’t consider it necessary to comment about it in the experimental section.
The headings on the top row in Table 8 are not written in English.
Indeed, they were corrected on the table.
Is the shape factor h for HM-EPT 1.3727 or 1,3727?
According to the writing standards of the International System SI, either a coma or a point can be used to indicate the decimal part of a number. Throughout the text we used the point for the decimal part, but unfortunately in table 8 there was a change in style and the coma was used. The whole decimal part has been changed to points.
Please provide references to support your explanation on “grain shape factor h of the metallographic grains at the edges of the specimens turned with HM-EPT are slightly higher than their counterparts turned without EPT, consequently reducing the number or grain edges or limits, which diminishes the susceptibility to intergranular corrosion.”
Variations in grain shape factors were obtained applying the Linear Intercept Procedure [34] and the condition of significant variation was determined by the ANOVA test. Authors such as Taiwade et al. [36], Jinghui Li et al. [37] and Yu et al [38] have reported inverse relationships between grain size and susceptibility to intergranular corrosion. Based on the results obtained, we the authors will continue the research to corroborate this statement.
In the abstract, authors claimed that “changes in corrosion behaviour can be explained in terms of metallographic micro hardness variation”. However, I do not find any relevant information explaining this mechanism in the discussion.
Thank you for the observation. Indeed, there was an error in the abstract. Line 18 was corrected, changing “metallographic microhardness” for “grain shape factor”.
Round 2
Reviewer 1 Report
In my opinion, the paper requires minor revision. In order to improve this work Authors should take into account the following comments:
Line 109-129: the equation (3) does not match with the equation cited in the reference [29] (page 489), please check the equations (3), (2), (1).
Line 166: according to international standard ISO 9227, NSS test should be conducted in 5 wt.% sodium chloride (NaCl) aqueous solution, not in sodium chlorite (NaClO2) which is completely different chemical compound. Thus, this sentence should be corrected.
Line 184-191: the calculation of the equivalent weight is incorrect, assuming that the chemical composition in Table 1 is in wt.%, EW is equal to c.a. 27.92. Moreover, Authors should take into account that according to standard G102 only the elements with concentration above 1 wt.% in the alloy (in this case only iron) should be considered in this calculation.
Line 198-199: the word “potentiostatic” means “at a constant potential”, so in these techniques, there can be no scan rate as a parameter. Scan rate should be declared in potentiodynamic techniques. Thus, this sentence should be corrected.
Line 206: Chemical corrosion is related with direct reaction of a metal with an environment. In other words, the corrosion is chemical if Ox and Red particles exchange electrons directly. Electrochemical corrosion is a redox process in which investigated metal has to be in contact with an electrolyte solution that serves as a medium for transport of ions. Authors investigate corrosion process in aqueous solutions of NaCl (regardless in what form - liquid in a cell or drops(fog) in the chamber). Thus, the investigated corrosion mechanism for the studied steel is electrochemical.
Line 242 (Tab.6) (now Line 274): the question in review 1 concerned differences in the corrosion rate for the same material subjected to the same treatment in the same environment, for example - HM after 0 h and 24 h has Vc = 2.97 and Vc = 12.21 mpy, why? Moreover, why Vc determined for HSS EPT after 24 h of immersion decreases – in contrast to all other samples?
Line 306-315: “apparent grain size” and “shape factor” are different parameters but in the article they are described by the same letter h. It should be corrected.
Author Response
Line 109-129: the equation (3) does not match with the equation cited in the reference [29] (page 489), please check the equations (3), (2), (1).
The classical mechanics of chip removal, according to Groover, indicate how to calculate the shear plane area Ac for an orthogonal cut. Since in the present study an oblique turning was applied, the equations used here aren’t the same, but they were generated based on those found in the reference.
Line 166: according to international standard ISO 9227, NSS test should be conducted in 5 wt.% sodium chloride (NaCl) aqueous solution, not in sodium chlorite (NaClO2) which is completely different chemical compound. Thus, this sentence should be corrected.
It seems like a typing error. The word has been changed.
Line 184-191: the calculation of the equivalent weight is incorrect, assuming that the chemical composition in Table 1 is in wt.%, EW is equal to c.a. 27.92. Moreover, Authors should take into account that according to standard G102 only the elements with concentration above 1 wt.% in the alloy (in this case only iron) should be considered in this calculation.
We agree with your observation and we thank you for the correction. Your observation was taken into account (line 183) and the value of EW was adjusted to 27.92.
Line 198-199: the word “potentiostatic” means “at a constant potential”, so in these techniques, there can be no scan rate as a parameter. Scan rate should be declared in potentiodynamic techniques. Thus, this sentence should be corrected.
We agree with your observation and we thank you for the correction. Your observation was taken into account (line 198) and the word “potentiostatic” was changed for “potentiodynamic”.
Line 206: Chemical corrosion is related with direct reaction of a metal with an environment. In other words, the corrosion is chemical if Ox and Red particles exchange electrons directly. Electrochemical corrosion is a redox process in which investigated metal has to be in contact with an electrolyte solution that serves as a medium for transport of ions. Authors investigate corrosion process in aqueous solutions of NaCl (regardless in what form - liquid in a cell or drops (fog) in the chamber). Thus, the investigated corrosion mechanism for the studied steel is electrochemical.
The text was carefully reviewed and, indeed, there was an error. The word “chemical” was changed for “electrochemical” in line 206.
Line 242 (Tab.6) (now Line 274): the question in review 1 concerned differences in the corrosion rate for the same material subjected to the same treatment in the same environment, for example - HM after 0 h and 24 h has Vc = 2.97 and Vc = 12.21 mpy, why?
In this case, between 0 h and 24 h, the formation of an oxide layer probably occurs on the metal surface, due to the severe attack by the corrosive agent.
Moreover, why Vc determined for HSS EPT after 24 h of immersion decreases – in contrast to all other samples?
It is possible that, before 24 h, the oxide layer is already formed on the metal surface. Somehow, this layer prevents that the corrosive agent continues its severe attack on the exposed surface.
Line 306-315: “apparent grain size” and “shape factor” are different parameters but in the article they are described by the same letter h. It should be corrected.
You are right. Indeed, we apply the Linear Intercept Procedure method and we determined the “shape grain factor”. In lines 306, 313 and 316, the word “shape grain factor” was introduced.
Reviewer 2 Report
Good Job!
Author Response
Thank You very much for your attention!
Reviewer 3 Report
Authors have addressed all my concerns.